# Convolutional Neural Networks for Lymphocyte Detection in Immunohistochemically Stained Whole-Slide Images

**Zaneta Swiderska-Chadaj**[1*]     **Hans Pinckaers**[1]     **Mart van Rijthoven**[1]

**Maschenka Balkenhol**[1]     **Margarita Melnikova**[1,3,4]     **Oscar Geessink**[1]

**Quirine Manson**[2]     **Geert Litjens**[1]     **Jeroen van der Laak**[1]     **Francesco Ciompi**[1]

[1]Diagnostic Image Analysis Group, Radboud University Medical Center, the Netherlands
[2]Department of Pathology, University Medical Center, Utrecht, the Netherlands
[3]Department of Clinical Medicine, Aarhus University, Denmark
[4]Institute of Pathology, Randers Regional Hospital, Denmark
[*]`zaneta.swiderska@gmail.com`

## Abstract

Recent advances in cancer immunotherapy have boosted the interest in the role played by the immune system in cancer treatment. In particular, the presence of tumor-infiltrating lymphocytes (TILs) have become a central research topic in oncology and pathology. Consequently, a method to automatically detect and quantify immune cells is of great interest. In this paper, we present a comparison of different deep learning (DL) techniques for the detection of lymphocytes in immunohistochemically stained (CD3 and CD8) slides of breast, prostate and colon cancer. The compared methods cover the state-of-the-art in object localization, classification and segmentation: Locality Sensitive Method (LSM), U-net, You Only Look Once (YOLO) and fully-convolutional networks (FCNN). A dataset with 109,841 annotated cells from 58 whole-slide images was used for this study. Overall, U-net and YOLO achieved the highest results, with an F1-score of 0.78 in regular tissue areas. U-net approach was more robust to biological and staining variability and could also handle staining and tissue artifacts.

## 1 Introduction

The immune system is a complex interplay of many differing types of cells, for example macrophages and lymphocytes, that work together to defend the body against bacteria, viruses, or tumor cells. In the last several years, the role of lymphocytes in the bodies own ability to fight cancer has become an area of great research interest after new, highly effective, immunotherapy strategies had been proposed [22, 16, 10]. Typically, the focus is on lymphocytes that occur within the tumor area, which are called tumor infiltrating lymphocytes (TILs). The hypothesis is that the balance between subsets of T-cells with pro- and anti-inflamatory function are important for disease progression [7]. The presence of TILs is related to patient prognosis after undergoing surgery or immunotherapy [4, 18, 20].

1st Conference on Medical Imaging with Deep Learning (MIDL 2018), Amsterdam, The Netherlands.

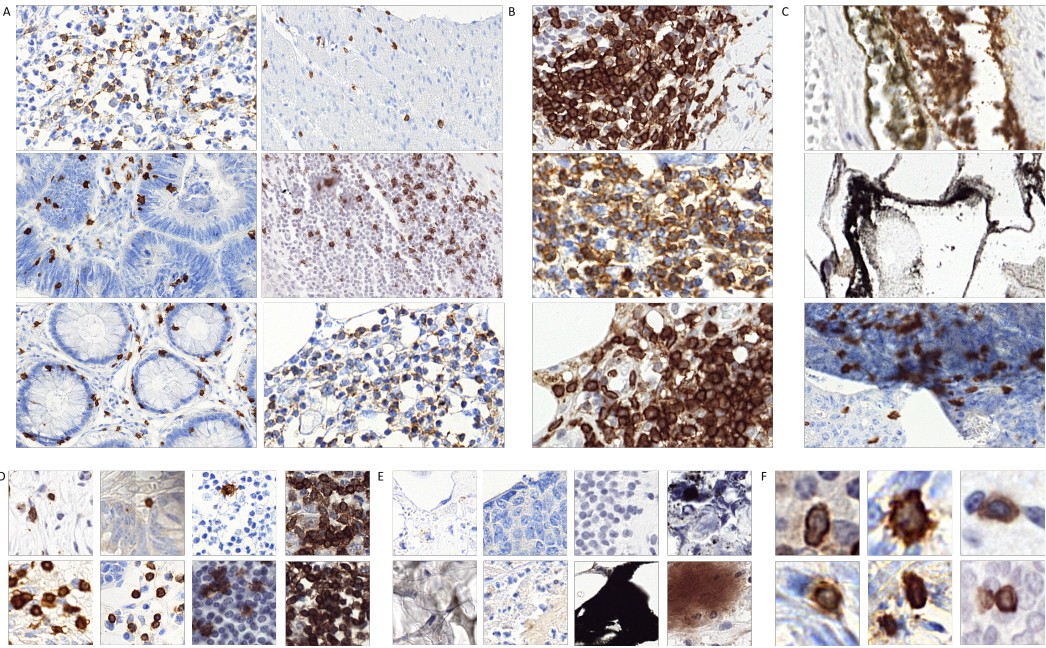

Figure 1: Example of part of Region of interests (ROIs) and extracted patches used for networks training, where: A- ROIs with regular tissue areas, B- ROIs with cluster cell areas, C- ROIs with artifacts or damaged tissue areas, D- patches with positive class include lymphocytes, E- patches with negative class include background, other cells, damages or artifacts, F- single lymphocytes.

Accurate detection and assessment of presence of lymphocytes in cancer could potentially allow for the design of new biomarkers that can help monitor the rapid progression [19] of a tumor. Moreover, automated tools to quantify the immune cell density and their localization in the proximity of tumor cells might help to predict the presence and development of metastases and overall survival of cancer patients.

Immunohistochemical staining (IHC) is a technique that allows to target specific cell types, including lymphocytes, by attaching a colored label to a specific antigen in (subcompartment of) a cell. In this way, immune cells can be distinguished from other type of cells. Widely used immune cell markers are CD3 (general T-cell marker) and CD8 (cytotoxic T-cell marker). Both are membrane markers, meaning that they target an antigen in the cells membrane, resulting in a brown-colored ring in positive cells (fig.1.F). It should be noted that stained specimens containing staining artifacts (i.e., dark brown areas) as well as damaged tissue regions are relatively frequent (see Fig. 1). On the research purpose, tissue areas were categorized into: (1) regular tissue areas- typical tissue areas without artifacts, damage or large areas of cell clusters; (2) cell cluster areas- areas include significant areas of clustered cells; (3) artifact areas- areas include various type of artifact, damages or ink blobs.

Manual assessment via light microscopy is the standard approach in research to detect and quantify lymphocytes. Given the very large amount of lymphocytes (≈100,000) in a single cancer tissue specimen, manual assessment at whole-slide image level is a very tedious, time-consuming, and therefore unfeasible task. Moreover, manual assessment suffers from intra- and inter- observer variability. Consequently, a method for automatic detection and quantification of immune cells is of great research and clinical interest.

The task of cell detection is a very popular topic in digital pathology. Computer-aided methods can significantly improve the objectivity and reproducibility of cell detection. Several approaches have been proposed for automatic cell detection on different types of digitized microscopical specimens and for various type of stained specimens [24, 8]. In many cases, detection algorithms are based on morphological operations, region growing, analysis of hand-crafted features and image classifications. However, these approaches usually avoid regions with artifacts [6].

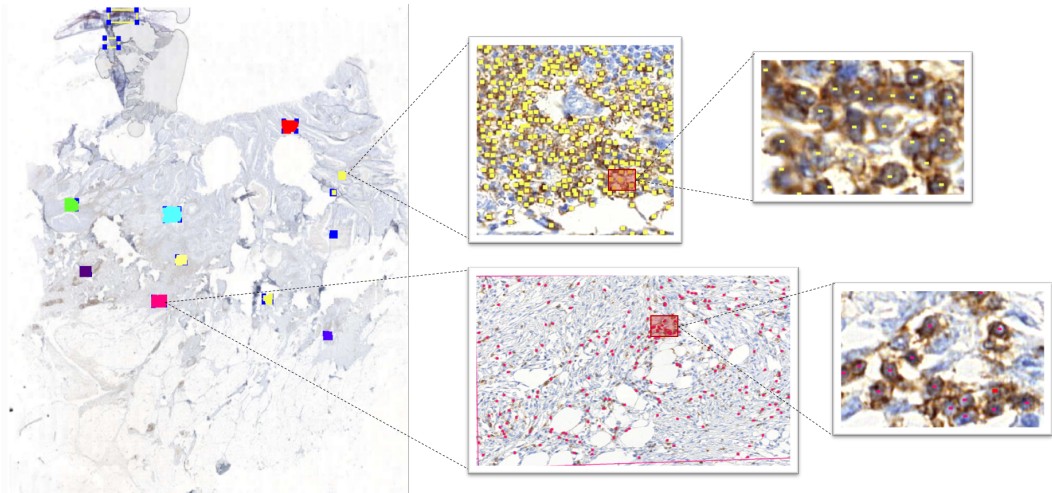

Figure 2: Example of WSI with annotations for selected ROIs.

The significant biological variability and influence of staining are challenges in digital cell detection, because of the large variability in appearance of stained cells (fig. 1.F). Deep Learning (DL), and in particular Convolutional Neural Networks (CNNs) can be a useful approach to tackle this kind of problem, because of their capability of automatically learning a multilevel hierarchy of image features [24]. Deep learning methods have been successfully used for nuclei identification on microscope images [25, 23, 17, 21]. Wang et al.[21] proposed a combination of hand-crafted features and CNN-based features to tackle the detection of mitotic figures. Xu et al.[25] proposed stacked sparse auto-encoder to learn a high-level representation of nuclear and non-nuclear objects. Xie et al.[23] proposed to compute a map of cell detection by using a structural regression approach based on CNN. Sirinukunwattana et al.[17] proposed deep learning approach sensitive to local neighborhood to detect and classify several types of nuclei in colon cancer specimen stained with H&E. A similar problem was tackled in [9], where an approach to lymphocyte detection in H&E was also proposed. However, different types of microscopy images or staining techniques produce a large variability in the appearance of tissue and cells. For this reason, robust methods for nucleus detection and segmentation are required.

In this paper, we address the problem of automatic detection of lymphocytes in whole-slide images of breast, colon and prostate cancer stained with CD3 and CD8 IHC stains. Motivated by recent advances in the field of computational pathology, we approach the problem using convolutional neural networks, and developed four different approaches that address the cell detection problem from different angles. We postulate that convolutional networks can learn to detect lymphocytes via (1) classification of patches centered on cells,(2) segmentation of cells, (3) detection of bounding boxes containing cells and (4) prediction of location of the center of a cell . Based on this assumptions, we developed methods based on (1) fully-convolutional networks (FCNN) [12], (2) U-Net [13], (3) You Only Look Once approach (YOLO) [14], and (4) Locality Sensitive Method (LSM) [17].

The four developed models were built using the same training and validation sets of 37 and 6 whole-slide images (WSIs), respectively, and finally tested on an independent data set of 15 WSIs containing 52,541 manually annotated lymphocytes.

The main goal and contribution of this work is to identify and compare efficient approaches for detection of lymphocytes in histopathology whole-slide images, with a particular focus on areas that are known to be difficult to analyze, such as regions containing stain artifacts. Finally, with this work we provide an overview of potential and limitations of several DL approaches applied to lymphocyte detection.

## 2   Materials

For this study, we collected 58 glass slides of breast, prostate and colon cancer specimens. These were immunohistochemically-stained for CD3 and CD8 from 6 different medical centers in the Netherlands, thus covering a wide range of staining protocols. These slides were subsequently digitized with a 3D-Histech Pannoramic Flash II scanner, resulting in whole-slide images with a resolution of 0.24 microns per pixel.

Lymphocytes in all slides were manually annotated within pre-selected regions-of-interest (ROIs) by experienced observers using the open-source ASAP software [11]. Within the regions of interest, lymphocytes were annotated exhaustively (see fig.2). ROIs were used as it is infeasible to perform exhaustive annotation at the whole-slide level, which contains millions of cells. ROIs were selected such that they cover all types of areas that occur in whole-slide images: regular tissue, clustered cells, and staining or tissue artifacts (see fig.1). For each slide, 6 to 14 ROIs were selected, where $\approx 60\%$ represents regular tissue areas and 40% areas with clustered cells or artifacts. After annotation this resulted in a total of 109,841 lymphocytes in 659 region of interests.

Subsequently, the 58 whole-slide images were divided into a training, validation and test set: 37 WSIs were used for training, 6 WSIs for validation, and 15 WSIs for testing. Training and validation slides were selected from two medical centres, whereas the independent set of test slides was created using data from all six centres. No intersection of patient data across sets was present. The validation set contained two images of each considered organ (breast, colon, prostate), one stained with CD3 and one stained with CD8. The test set contained five images per organ, with a roughly even proportion of CD3 and CD8.

## 3   Methods

Four independent approaches for detection of lymphocytes, based on supervised convolutional networks were developed, namely: (1) patch classification using fully convolution neural network (FCNN), (2) semantic segmentation using U-net, (3) bounding box detection using the You Only Look Once (YOLO) network, and (4) prediction of cell center locations by locality sensitive networks (LSM). All methods used exactly the same training and validations slides, and most of them are based on CNN patch classification followed by post-processing. Methods were developed using both Tensorflow [1] and Keras [3], or Theano [2] and Lasagne [5]. All networks were evaluated in the same way and individually optimized to obtain the best possible F1-score on the validation set. A detailed description of each method is provided in this section and the optimal set of parameters for each network is presented in the results section.

### 3.1   Learning to "classify"

Lymphocyte detection can be formulated as a binary pixel-wise classification approach, where the two classes are (1) lymphocytes and (2) other structures. CNNs are then trained to classify patches as either class 1 or 2. To allow efficient processing of regions of interest after training, fully-convolutional networks [12] can be used. In such a network, all layers are implemented as convolutional layers. Processing an ROI with a trained fully-convolutional network results in a likelihood map, where each pixel represents the likelihood of a lymphocyte being present at that location.

The architecture we used in this paper contains six convolutional layers. The first two layers are interleaved with pooling layers to improve spatial invariance. Before the last two layers dropout is applied. Every convolutional layer is followed by a rectified linear unit, except the last layer where a softmax nonlinearity is applied. To prevent loss of resolution due to the pooling layers, upsampling to a pixel-level classification is performed via shift-and-stitch implemented using filter dilation.

Last, likelihood maps are post-processed based on a Gaussian filtering ($\sigma$=4) and a regional maxima operation applied on the detection mask. The regional maxima operation is performed on the image after application of the Gaussian filter, with neighborhood parameter equal 15. This results in a list of detected lymphocyte locations.

## 3.2 Learning to "segment"

Lymphocyte detection can also be formulated as a semantic segmentation problem. In the context of semantic segmentation in medical imaging, the U-net architecture [13] has been widely used. The model includes a contracting path to capture image context and a symmetric expanding path that enables precise object localization.

In our study, we extended the original architecture by adding dropout layers between convolutional layers, with the aim of reducing overfitting.

In order to train U-net, target segmentation masks are required. For this reason, manually annotated dots were converted into elliptic regions, where the radii of the ellipses was defined empirically. To better deal with cells in close proximity, we defined target masks that included four classes: cell border, cell body, cell center and background class. The *center* class is a small circle around the manually annotated point, while the *body* and the *border* classes are a ring with elliptical shape (see Fig. 3).

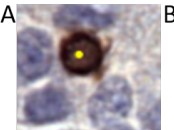 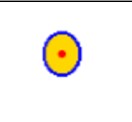

Figure 3: Example of a elliptical mask (B) derived from an annotated cell (A).

The output of the U-net network is post-processed in order to obtain the final detection of lymphocytes. Post-processing is based on a Gaussian filtering ($\sigma$=4) and a regional maxima operation applied to cell centers detection mask. The regional maxima operation is performed on the image after application of the Gaussian filter, with neighborhood parameter equal 15. As a result, a map of detected lymphocytes is created.

## 3.3 Learning to "detect"

Object instance detection has been a hot topic in the computer vision community, with many competing approach. In this paper, we chose the YOLO (You Only Look Once) architecture as the representative of this class of methods [14, 15]. YOLO processes an image by first dividing it into a grid, and then predicting bounding boxes and classes for each grid cell. In this paper, YOLO is trained to predict bounding boxes that contain lymphocytes. Each center coordinate of a target box, that falls within a grid cell, will make that particular grid cell responsible for detecting the target lymphocyte. For each bounding box, the network outputs a confidence level $C$, which is a measure of how sure the model is that a lymphocyte is captured by that bounding box, instead of background.

For training purpose, we used follow the loss function presented in [15], and a prediction was considered as correct if the predicted bounding box had an intersection over union (IoU) $\geq 0.5$ with a true bounding box. During inference, predicted bounding boxes with overlap are considered as detecting the same lymphocyte. Therefore, non maximum suppression is applied in order to keep only one prediction at test time.

## 3.4 Learning to "localize"

Lymphocyte detection can also be addressed by predicting the location of the center of a lymphocyte in a small patch. This approach is based on the *locality sensitive method* (LSM) presented in [17], where it was applied to cell localization in H&E images.

The goal of the method is to learn a mapping function $\mathcal{F}$ from a 2D input domain (i.e., an image patch) $I(x, y)$ to $M$ output locations $(x_m, y_m)$, which correspond to locations of centers of lymphocytes, as well as a likelihood parameter $h_m$ for each location: $\{x_m, y_m, h_m\} = \mathcal{F}(I(x, y))$, for $m = 1, \ldots, M$.

In this way, $M$ represents the *maximum* number of lymphocytes that are expected to be found in an input patch $I(x, y)$, and the output variable $h_m$ allows to control the likelihood of each predicted location to contain an actual lymphocytes. If a patch contains $L$ lymphocytes, and $L < M$, then some values of $h_m$ will be $\approx 0$. In this paper, we trained the same architecture used in the original paper [17] and we implemented a *spatially constrained layer* that, differently from [17], predicts an output map which has the same size as the input patch.

The output of the network is a 2D map of Gaussian-like profiles that indicate the predicted locations of lymphocytes. In order to improve the robustness of predictions, as in [17] we process each input

Table 1: Network parameters, where: SGD- stochastic gradient descent, CCR- categorical crossentropy, BCR- binary crossentropy, NM- Nesterov Momentum, px-pixels.

| | learning rate | Dropout factor | Optimizer | loss function | nb of epochs | batch size | resolution | input size(px) | output size(px) |
|---|---|---|---|---|---|---|---|---|---|
| FCNN | 0.00005 | 0.5 | ADAM | CCR | 150 | 64 | x40 | 64x64 | 1x1 |
| U-net | 0.05 | 0.5 | SGD | CCR | 10 | 1 | x20 | 128x128 | 128x128 |
| YOLO | 0.00005 | - | ADAM | yolo loss | 200 | 4 | x20 | 256x256 | - |
| LSM | 0.05 | - | NM | BCR | 1000 | 32 | x20 | 27x27 | 27x27 |

patch multiple times and accumulate the predicted profiles. For this purpose, we shifted the patch $n_s = 4$ times per image dimension, therefore cumulating $(n_s + 1)^2 = 25$ predictions per patch. The final set of locations $(x, y)$ of predicted lymphocytes is extracted by detecting local maxima on thresholded prediction maps.

## 4 Validation

### 4.1 Method parameters

Due to the different nature of the four considered approaches, each network architecture and corresponding hyper-parameters were optimized independently. In several cases, model settings such as network architectures, dropout values, learning rates, were initially set as proposed in previous works, and successively fine-tuned based on the problem at hand. An overview of the obtained, optimal, hyper-parameters is provided in Table 1.

Some method-specific parameters were also optimized. For the FCNN, the training patch size was set as a 32x32 pixels. For YOLO, the bounding box size was based on the average size of a lymphocyte (6-8 µm) and was set to 12 pixels. For the loss function, $\lambda_{coord} = 5$, $\lambda_{noobj} = 1$. For effective elimination of unsure and redundant boxes, thresholds for object confidence and non maximum suppression were set to 0.2 and 0.1, respectively. For the LSM approach the parameters of the spatial constraints layer were set to $d = 4$ and $M = 2$.

### 4.2 Experiments

In this section, we present the results of conducted experiments, which include: (I) comparison of various DL approaches for lymphocyte detection, (II) analysis of the influence of the training dataset on the results, (III) comparison of automatic lymphocyte detection with manual annotations on the independent test set.

The reference standard consists in sets of dots annotated inside of cluster of cells (fig.2). In order to define a *hit criterion*, we considered a valid region within a radius of $r=16$ pixels from the annotated center of each nucleus as a ground truth. The $r$ value was established based on a typical lymphocyte radius, which is in a range 3-4 µm, (13-18 pixels at 40X magnification, with resolution of 0.228 µm/pixel). In practice, it means that the distance between manual annotated cell and detected cell should not be higher than the average value of lymphocyte radius.

If detected cell is within a valid ground truth area, then it is counted as a true positive, in other cases it is counted as a false positive. If ground truth area is not detected, then a detection is counted as false negative. Based on these criteria, the following metrics were computed: Precision, Recall, F1-score and the FROC curve.

#### 4.2.1 Comparison of various DL approaches for lymphocyte detection

The objective of this experiment is to detect all lymphocyte nucleus in ROIs by locating their center position using each of the developed methods presented in section 3. A comparison performed on the validation dataset (6 WSIs) included three types of region: (a) regular tissue areas, (b) areas containing clustered cells and (c) areas with staining or tissue artifacts (Fig.1).

Quantitative results are presented in Table 2 and in Fig. 4. The F1-score is in the range of 0.66-0.80 for regular tissue areas. The FCNN is the simplest approach and also achieved low F1-score value equal 0.63. Highest performance was achieved by U-Net and YOLO with 0.75 and 0.72 respectively.

In the presence of clusters of cells, F1-score was in the range of 0.65-0.75, while in the presence of artifacts, lower F1-score values were observed, in the range of 0.40-0.64. The best result for all analyzed regions was achieved with the U-net approach (F1-score of 0.75). It should be noted that results for YOLO and U-net approaches are similar for regular ROIs and ROIs with clustered cells. The U-net approach seems to work considerably better than the other approaches in the presence of artifacts (F1-score of 0.64 vs. 0.49 for the second best approach).

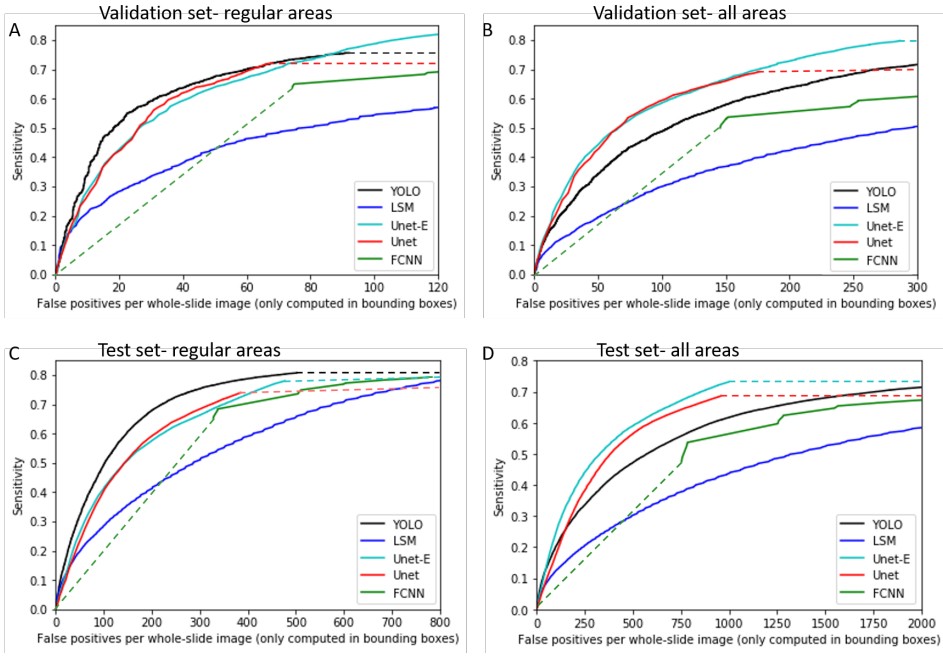

Figure 4: FROC-curves for DL models for the validation set (A, B) and test set (C, D), where: A- the FROC-curve for regular tissue areas in validation set, B- the FROC-curve for all tissue areas in the validation set, C - the FROC-curve for regular tissue areas in the test set, D - the FROC-curve for all tissue areas in the test set, green- FCNN, blue- LSM, black-YOLO, red- Unet, cyjan- Unet-E, dashed line- interpolation.

### 4.2.2 Influence of the training dataset

Initial visual inspection of the results on the validation set showed that all methods performed much more poorly in areas with clustered cells and artifacts. We hypothesized that this is partly caused by sampling too few difficult examples from these areas at training time. This prevents the networks from adequately learning how to deal with difficult areas. To test this hypothesis we set up an experiment in which we only sample patches from difficult areas, for example cell clusters for the positive class and areas with artifacts for the negative class. As this limits the number of samples we can generate we also applied simple data augmentation including oblique rotation. The final 'hard' training set contained 32,684 patches, which is still smaller than 'basic' training set.

The U-net based approach, which gave the highest F1-scores in Experiment I, was used for this experiment. It was trained *de-novo* with the 'hard' training dataset. The parameters and post-processing was not modified. The results for this approach are described as Unet-E in Table 2 and Fig.4. The F1-score increased with 0.029 overall and, interestingly, also for the easy ROIs (increase in F1-score from 0.79 to 0.83).

Table 2: Statistical results for the validation set and the independent test set. The bold values represent the maximum of F1-score for each of area.

| Area type | Method | Validation set | | | Test set | | |
|---|---|---|---|---|---|---|---|
| | | F1-score | Precision | Recall | F1-score | Precision | Recall |
| Regular tissue | FCNN | 0.727 | 0.741 | 0.713 | 0.721 | 0.753 | 0.810 |
| | LSM | 0.659 | 0.674 | 0.644 | 0.669 | 0.554 | 0.846 |
| | YOLO | 0.802 | 0.853 | 0.757 | **0.780** | 0.750 | 0.810 |
| | Unet | 0.794 | 0.884 | 0.720 | 0.762 | 0.785 | 0.740 |
| | Unet-E | **0.825** | 0.827 | 0.821 | 0.778 | 0.756 | 0.781 |
| Cell clusters | FCNN | 0.651 | 0.698 | 0.609 | 0.692 | 0.771 | 0.623 |
| | LSM | 0.680 | 0.707 | 0.656 | 0.665 | 0.730 | 0.610 |
| | YOLO | 0.748 | 0.823 | 0.685 | **0.763** | 0.820 | 0.710 |
| | Unet | 0.720 | 0.803 | 0.640 | 0.737 | 0.836 | 0.659 |
| | Unet-E | **0.758** | 0.751 | 0.764 | 0.754 | 0.799 | 0.714 |
| Artifact or damage areas | FCNN | 0.528 | 0.380 | 0.864 | 0.271 | 0.164 | 0.820 |
| | LSM | 0.403 | 0.289 | 0.673 | 0.170 | 0.094 | 0.880 |
| | YOLO | 0.486 | 0.351 | 0.790 | 0.250 | 0.146 | 0.850 |
| | Unet | 0.638 | 0.616 | 0.661 | 0.422 | 0.309 | 0.669 |
| | Unet-E | **0.656** | 0.570 | 0.770 | **0.490** | 0.374 | 0.710 |
| All areas | FCNN | 0.630 | 0.638 | 0.623 | 0.598 | 0.544 | 0.672 |
| | LSM | 0.598 | 0.587 | 0.611 | 0.488 | 0.371 | 0.710 |
| | YOLO | 0.723 | 0.717 | 0.730 | 0.607 | 0.520 | 0.730 |
| | Unet | 0.750 | 0.820 | 0.690 | 0.700 | 0.715 | 0.686 |
| | Unet-E | **0.779** | 0.761 | 0.799 | **0.725** | 0.719 | 0.732 |

### 4.2.3 Comparison on the independent test set

For the last experiment we took all four, fully optimized, methods and applied them to the independent test set, which contains 15 whole-slide images from all six institutions and all three organs (breast, prostate, colon). The test data thus also includes centers which were not included in the training set.

In total, 166 ROIs in test data were exhaustively annotated, resulting in 52,541 cells in the test set. The ROIs have varied from $0.2mm^2$ to $23.8mm^2$ (on average $12mm^2$).

Table 2 and Figs. 5 and 6 present results for each of the considered approaches, showing that Unet-E and YOLO allow for accurate cell detection in the independent test set. The F1-score is equal 0.78 for regular tissue areas, and it ranged from 0.61-0.73 across all ROI types. Relatively high F1-scores were also obtained for difficult areas such as cell clusters (highest F1-score of 0.76) and artifacts (highest F1-score of 0.49).

### 4.3 Discussion

In this work, we compared four deep learning based methods for automatic detection of lymphocytes in a large multi-center cohort of immunohistochemically-stained tissue of breast, colon and prostate tumors. Our results show that all methods can achieve decent results in terms of F1-score, with the lowest score being 0.49 for the locality sensitive network.

Nonetheless, significant differences between the methods can be observed. The overall best results were obtained with the U-net approach trained on a *'hard'* training set, with an overall F1-score of 0.73 on the test set. The second best method, which is the YOLO instance detection network, is very close to the U-net approach, except in the case of staining or tissue artifacts. In those ROIs the differences between YOLO and the best U-net is 0.24 in terms of F1-score, which is a big gap.

A key strength of this study is the use of data in the test set from six different medical centers, each with their own staining protocol. This makes it possible to assess the robustness of algorithms to different protocols. We can for example see that YOLO shows a significant drop in performance from validation to test set, indicating that it is either not robust to data from different centers or has trouble generalizing to unseen data. The same holds for the LSM algorithm. FCNN and U-net seem to be more robust and show a smaller drop in F1-score from validation to test set.

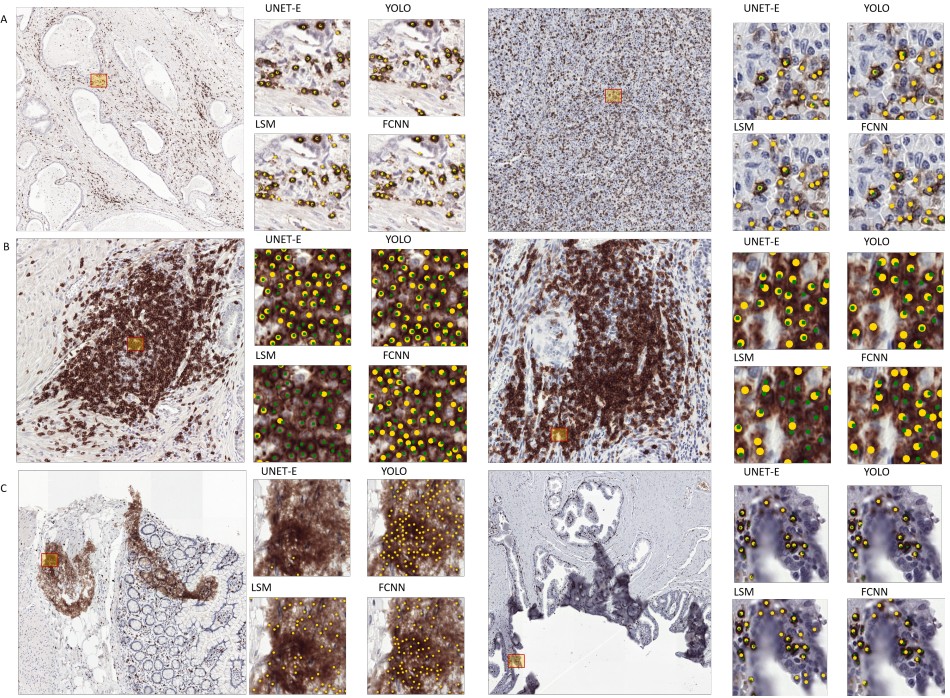

Figure 5: Presentation of the deep learning methods' performance on the test slides, where: yellow-detected cells, green- manually annotated cells.

The application of the LSM method to lymphocyte detection in immunohistochemically-stained slides shows the poorest results across the four methods. Especially in areas containing artifacts the performance is abysmal with an F1-score of 0.17, which makes it unusable for those areas. (see fig.5). The LSM network in this paper was a direct reimplementation of the one that was used for H&E images, where artifacts are not so much present. This could potentially be addressed by increasing the capacity of the network or by modifying the field-of-view in combination with the parameter $M$ for the number of detected cells.

The FCNN has the third best results of the four methods. It is easy to train and fast, however, it has one major flaw compared to U-net or YOLO and that is the single output label per patch. The labels in YOLO and U-net contain much more information, allowing the networks to handle, for example, multiple positive cells per input patch natively.

The YOLO method achieved one of the highest results in the cell detections in regular and cell cluster areas (see fig.5). The advantage of YOLO is taking the global information from a sampled patch into account and directly optimizes detection performance calculated from grid cells. This approach reasonably predicts lymphocytes in WSIs. However, this plain implementation of YOLO can not deal with artifacts. Using the 'hard' training set for YOLO as well might solve this limitations, but solutions could potentially also be found in the definition of the loss function.

The semantic segmentation (U-net approach) achieves good results in regular areas and cell cluster and the best results in areas containing artifacts. The application of the three-class cell masks makes is easier to separate touching cells and accurately localize the cell center, which is impossible in a case of binary masks. In addition, it is the best method by far on areas containing artifacts, even without training on the 'hard' data set.

In addition, we show that the performance of the U-net can be further improved by specifically training it with difficult samples. Perhaps unexpectedly, the performance increase in cell clusters and areas with artifacts is accompanied by an increase in the F1-score in the regular areas as well. In future work we will more extensively assess the effect of the composition of the training data set on the performance of the other methods as well.

A key strength of this paper is the inclusion of areas with cell clusters and artifacts specifically. In literature, most authors focus only on cell detection in regular tissue areas. This tends to give an overly optimistic result, and makes it hard to discriminate between methods. To give an example, the worst and best method on regular tissue areas are separated 0.11 in F1-score. In areas with artifacts this is 0.32.

Although we evaluated the different deep learning methods in ROIs to make the study feasible, all presented methods can be applied at the whole-slide level, see Figure 6 in the supplemented materials. In this example, 78,346 positive cells are detected in a single whole-slide image. This already shows the key advantage of automated analysis of whole-slide images, counting these cell manually would be completely infeasible across any meaningfully sized data set.

In this study we specifically looked at CD3- and CD8-positive immune cells. However, many more stains are used to describe the immune system, both with similar expression patterns, such as CD4, but also with different expression patterns, such as CD45RO which marks memory T-cells. In future work we would like to evaluate our methods on these staining as well, ideally using a single method to completely analyze the entire immune response in the tumor environment.

Summarizing, we have shown that deep learning techniques can be applied to detect positively stained cells in immunohistochemistry, with great promise for immuno-oncology. The fact that we can now reliably quantify these cells opens an avenue of research in which we relate immune cell quantities to tumor progression and treatment response.

### Acknowledgments

We thank Sophie van den Broek and Dennis Otte for support in the process of immune cell annotations.

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

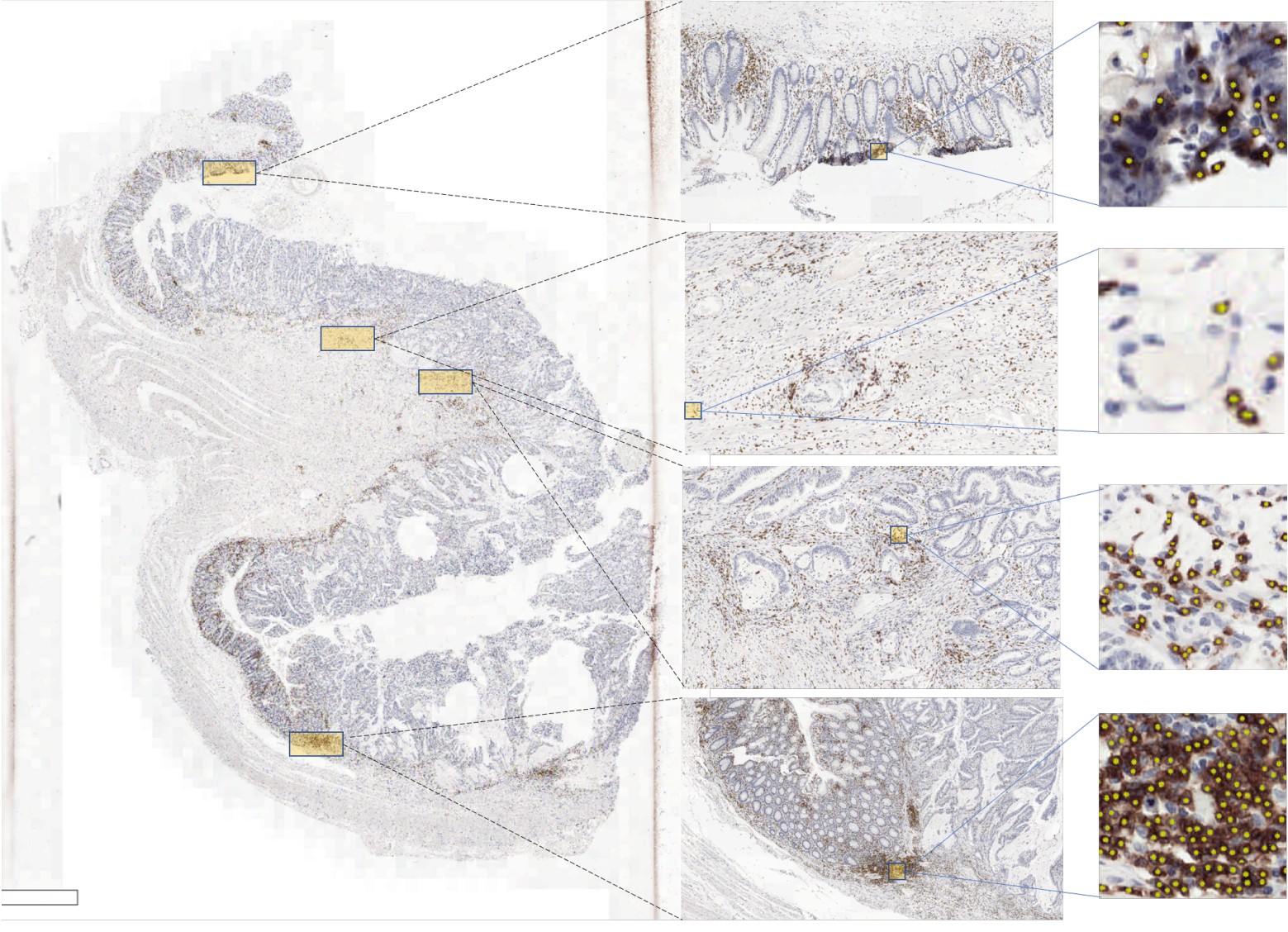

Figure 6: Example of lymphocyte detection on the whole slide image with zooming of selected areas.