# OpenReview forum: "Convolutional Neural Networks for Lymphocyte detection in Immunohistochemically Stained Whole-Slide Images"
_MIDL.amsterdam/2018/Conference — MIDL 2018 Poster_

### Review · AnonReviewer1 · 2018-05-09
**Nice way to use CNNs for lymphocyte diagnosis**

**Rating:** 3
**Confidence:** 2

**Review:**

Summary:
The authors investigated the use of convolutional neural networks (CNN) for the automated diagnosis of Lymphocytes in immunohistochemically stained whole-slide images. This paper presents a comparative evaluation of the performances of several state-of-the-art methods. The authors conclude that U-net performs best on this task.

Pros:
* Paper is well written. Difficulty of problem, purpose of this paper, materials, methods, results, discussions are well introduced.
* Segmentation, detection, and localizations can be conducted with the tested methods (note that each work is conducted individually). This work can assist entire diagnosis of cancer.

Cons:
* This work is mainly focusing on comparing between existing network structures. It is important work but there is little novelty from a technical point of view.
* Did it make sense to introduce dropout layers to U-net? Comparison between w/ and w/o dropout is recommended.

Overall opinion:
* Despite the lack of technical novelty, the paper does provide a comprehensive analysis of current state-of-the-art methods for detect positively stained cells in immunohistochemistry, which could be promising for immuno-oncology and be of interest for the medical imaging community.

Specific comments:
* Section 1 is long without divisions. Please divide into Introduction, Related Work, and Purpose.
 * Make Sec. 4.3 as individual section (Sec. 5), and divide into subsections for each topic
* Title: detection --> Detection
* Sec 2: Insert image size [pixel]
* Sec 4.1: \lambda_{coord} ... Italic "coord" means the product of scalars c, o, o, r, and d. noobj as well
* Sec 4.2: Do not make "=16 px" italic. px means pixel?
* Sec 4.2: 0,228 --> 0.228
* Table 2: What does bold values mean?
It seems not maximum, e.g. 0.810 in Regular-tissue, Test set, Recall.
* Sec 4.3: 78.346 --> 78,346
* Sec 4.3: deep Learning --> deep learning
* Fill citation [11].
* Figure 6: Make the caption more detailed.


**Special Issue:**

No

---

### Review · AnonReviewer3 · 2018-05-09
**Comparison of 4 standard architectures for lymphocyte detection**

**Rating:** 2
**Confidence:** 2

**Review:**

The paper presents a detailed evaluation of 4 standard architectures applied to the problem of lymphocyte detection from immunohistochemically stained images. The dataset is carefully selected, including glass slides of different cancers (breast, prostate and colon cancer specimens), 6 different centers, and two different types of staining. I have never worked with digital pathology myself and cannot comment on the novelty of this evaluation. I would expect this study to be of interest for all researchers working on a very similar application, but due to different design choices in the different approaches, it is hard to derive conclusions that might generalize to other problems.

It is interesting to see that training on a smaller number of difficult cases improves results on all regions.

The selection of the 4 different approaches compared appears a little arbitrary. It is also unclear whether the differences in performance are due to differences in the type of approach or rather differences in details of the architectures (e.g. number of layers) or the applied post processing. For instance, U-Net performs much better than FCN. Is that due to the expanding path of the Unet, which is the essential difference between the two, or due to differences in number of layers etc between the two implementations, or the fact that the U-net approach in these experiments is allowed to use information of cell border and body classes while the FCN is not? It is mentioned in the discussion that the additional classes improve performance, but this is not shown. As another example, Yolo does not generalize well between validation and test data compared to Unet. Is this problem inherent to the YOLO architecture or could it be influenced by other factors (e.g. it seems the specific process, grid-wise application for YOLO, and the post processing done for Unet could be important)?

The method descriptions are quite brief and at times confusing. It would be helpful to include a bit more detail on the different architectures to make the paper self-contained. E.g. 3.1 – Is this the same architecture from [12]? what is the size and number of convolution filters in the different layers? What type of pooling is used and where (“the first two layers are interleaved with pooling layers” – does it mean there is a single pooling layer between convolution layers 1 and 2? ) how are ellipse radii in 3.2. “determined empirically”? What is the YOLO loss function?

Minor comments:
* 3.1 seems to be (judging from the first sentence and the upsampling by shift-an-stitch) about pixel-wise classification (which would turn “learning to classify” into “learning to segment” as well as “learning to localize”). However, later on it is stated as the major problem of FCN that it provides only a single label per patch and therefore cannot handle multiple cells per patch. Please clarify.
* Fig 3 caption – A and B swapped.
* Unet-E is shown in the figure described in 4.2.1. but explained only later.

**Special Issue:**

No

---

### Review · AnonReviewer2 · 2018-05-11
**Interesting paper about a comparison of different types of object detection methods.**

**Rating:** 3
**Confidence:** 2

**Review:**

This work compares four deep learning methods for the detection of lymphocytes in histology data. Lymphocytes detection and segmentation is a very relevant application in the medical imaging because of the possible use in cancer research. The paper is well written and easy to read.
The authors discuss and evaluate four different types of methods (“classification”, “segmentation”, “detection”, and “localization”), which I believe is very interested for the medical imaging community. Readers of this article will learn about the different ways to tackle an object detection problem and will also be educated on recent methods for each type of method. The authors also show the effect of training on harder data (data with artifacts and clusters of lymphocytes) with the segmentation method (U-Net). There it clearly improves the performance. It’s unclear why the authors didn’t run the same experiment for the other three methods. That would have been better. I’m now wondering if the detection method (YOLO) trained with harder data would have outperformed the U-Net with harder data.
Another point of improvement is adding more explanation about the differences and similarities between the different methods. The four methods are now explained independently, but for full appreciation of the methods it would have been great if the authors could have focused on how the methods exactly relate to each other. This especially holds for the classification versus the segmentation method and the detection versus the localization method.

**Special Issue:**

No

---

### Decision · Program_Chairs · 2018-05-15
**Paper36 Acceptance Decision**

Poster